# Numerical Model and Experimental Validation for Laser Sinterable Semi-Crystalline Polymer: Shrinkage and Warping

**DOI:** 10.3390/polym12061373

**Published:** 2020-06-18

**Authors:** Jiang Li, Shangqin Yuan, Jihong Zhu, Shaoying Li, Weihong Zhang

**Affiliations:** 1State IJR Center of Aerospace Design and Additive Manufacturing, School of Mechanical Engineering, Northwestern Polytechnical University, Xi’an 710072, China; lijiang235@mail.nwpu.edu.cn (J.L.); lishaoying@mail.nwpu.edu.cn (S.L.); zhangwh@nwpu.edu.cn (W.Z.); 2MIIT Lab of Metal Additive Manufacturing and Innovative Design, NPU-QMUL Joint Research Institute, Northwestern Polytechnical University, Xi’an 710072, China; 3Unmanned System Research Institute, Northwestern Polytechnical University, Xi’an 710072, China

**Keywords:** powder bed fusion process, thermo-mechanical model, shrinkage, warping, recrystallization-induced strain

## Abstract

Shrinkage and warping of additive manufacturing (AM) parts are two critical issues that adversely influence the dimensional accuracy especially in powder bed fusion processes such as selective laser sintering (SLS). Powder fusion, material solidification, and recrystallization are the key stages causing volumetric changes of polymeric materials during the abrupt heating–cooling process. In this work, the mechanisms of shrinkage and warping of semi-crystalline polyamide (PA) 12 in SLS are well investigated. Heat-transfer and thermo-mechanical models are established to predict the process-dependent shrinkage and warping. The influence of raw material- and laser-related parameters are considered in the heat-transfer and thermo-mechanical models. Such models are established considering the natural thermal gradient and dynamic recrystallization, which induce internal strain and volumetric change. Moreover, an experimental design via orthogonal approach is introduced to validate the feasibility and accuracy of the proposed models. Finally, the quantitative relationships of process parameters with product shrinkage and warping are established; the dimensional accuracy in part-scale can be well predicted and validated with printed parts in a real experiment.

## 1. Introduction

Powder bed fusion including selective laser sintering (SLS) is an essential category of multiple additive manufacturing (AM) technologies that is used to fabricate three-dimensional objects with arbitrary geometry via a digitally controlled layer-by-layer fusion-solidification process. Laser-sintered parts suffer unexpected dimensional instability due to the fact that the feedstock materials of semi-crystalline polymers experience significant volumetric change and complex phase transitions in SLS process. Therefore, the precise prediction and good control of dimensional accuracy become critical challenges. Performances of SLS components, particularly the shrinkage and warping, which are the primary observations of dimensional instability, are simultaneously influenced by variations of process parameters, including material-related variables, laser-related variables, and process conditions. In particular, laser power, slice thickness, scanning speed, hatching space, and temperature history are critical factors. In the traditional approach, these variables were routinely adjusted by the trial and error method, which is expensive, time-consuming, and unreliable, especially for a new material [1]. To overcome these shortcomings and to obtain the desired performance, the digital twin technology (DTT) for AM was explored through a digital replica (virtual model) evaluation as compared with its actual physical model. DTT for AM can be used to search optimal set of parameters and guide AM machine to automatically and adaptively change the process parameters [2]. This technology offers a digitalized feedback control for producing physical models as well as to retain the dimensional accuracy of their original virtual models.

Shrinkage is one of the prime reasons for part inaccuracy, particularly during the powder bed fusion process. Several attempts have been made to improve the accuracy of the AM parts by controlling the effect of shrinkage. Negi and Sharma [3] studied the influence of bed temperature, hatching space, laser power, scanning length, and scanning speed on the shrinkage of glass fiber/polyamide (PA)12 composite. They revealed that the bed temperature, hatching spacing, and scanning speed were the three primary factors influencing the shrinkage. Soe et al. [4] discovered that non-linear shrinkage always occurs, and the nonlinearity of shrinkage can be reduced by placing the components at the center of the working chamber. Verbelen et al. [5] evaluated the SLS processability of the materials PA12, PA11, and PA6. They discovered that shrinkage mainly depends on the cooling rate and crystallinity. Benedetti et al. [6] identified the main factors inducing shrinkage, which included thermal retraction, recrystallization, and powder densification. These efforts towards the investigation of shrinkage were mostly carried out according to experimental methods, which is costly and time-consuming. Additionally, the influence of individual factors such as recrystallization is impossible to evaluate, and the underlying mechanisms of shrinkage and warping are poorly understood. Thus, to investigate the physical mechanism of process-induced shrinkage and to accurately predict it, it is necessary to employ the integration of numerical modeling and experimental evaluation, showing the advantage of DDT in fitting the virtual and physical models.

To date, multi-scale and multi-physics numerical models have been investigated to provide insights into the mechanisms of printing processes and predict the properties of AM parts. The models are classified into three categories according to the modeling scale: micro-scale, meso-scale, and macro-scale. The micro-scale model was aimed at calculating the grain growth and solidification microstructure, and the meso-scale model was adaptive to predict the fluid flow of melting pool and evolution of individual powder particles [7,8,9,10]. The macro-scale model based on the finite element method (FEM) has been employed to resolve the product-scale process evolution, which is appropriately adopted to predict the shrinkage and warping in this work. Therein, the temperature profile is predicted by the heat-transfer model and the residual stress distortion is determined by the mechanical model. Zeng et al. [11] reviewed the heat-transfer analysis via SLS, including analytical solutions, numerical solutions, and thermal measurements. They conducted a case study to predict the temperature profile of laser scanning. Parry et al. [12] built a simple thermo-mechanical model to determine the temperature gradient-induced residual stress in the selective laser melting (SLM) process. The alternating and unidirectional strategies were performed to achieve a uniform plastic strain distribution upon the laser melting process. Li et al. [13] developed a thermo-mechanical model considering the temperature-dependent material properties, and found that the residual stress along the scanning direction was relatively large. Tan et al. [14] built a thermo-mechanical model including the phase transformation of powder–liquid–solid state. They found that the phase transformation leads to an increase of compressive stress and a decrease in tensile stress. However, many studies of macro-scale models are based on metallic materials such as Ti_6_Al_4_V, rather than semi-crystalline polymer materials, which have been used for a wide spectrum of applications in the SLS process. Meanwhile, these simulations have rarely been combined with corresponding systematical experiments. Additionally, for the numerical calculation of shrinkage/warping of the semi-crystalline polymer, the complete shrinkage model including thermal retraction, recrystallization, and powder densification has not been fully considered. Therefore, it is still challenging to search the optimal combination of parameters to minimize the corresponding shrinkage/warping and the specific quantitative relationship from material characterization to the part-scale shrinkage and warping, systematically coupling experimental works with modeling prediction. This eventually connects the virtual and physical model of DTT in practical terms.

In this work, the PA12 is adopted as a representative semi-crystalline polymer that exhibits typical shrinkage and warping effects upon abrupt temperature changes in the SLS process. A theoretical model combined with heat-transfer, thermo-mechanical, and material crystallization kinetics is developed to anticipate temperature distribution, residual stress, and entire shrinkage/warping at part-scale. The thermal analysis is conducted using a volumetric heat source to describe the scattering and absorption of laser energy. Then, the temperature distribution and melting pool are determined. Thereby the effective energy density window is obtained according to the fully sintering conditions. With the thermal, mechanical properties, and crystallinity characteristics of the polymer, the elasto-plastic constitutive law including thermal strain and recrystallization-induced strain is employed to evaluate residual stress, shrinkage and warping. Meanwhile, the predictions of shrinkage and warping are adequately validated by orthogonal approach. The quantitative correlations between shrinkage/warping and process parameters including laser power, slice thickness, scanning speed, hatching space, and “layer-layer angle” are also established. Thus, the shrinkage, warping, and their corresponding optimal-parameter sets can be achieved. Finally, a representative structure is employed for macro-scale shrinkage and warping prediction based on the proposed thermo-mechanical model to manifest its applicability.

## 2. Modeling and Experiment

### 2.1. Heat-Transfer Model

The three-dimensional heat-transfer model is described by the following equation:(1)ρCp⋅∂T/∂t=∇⋅(k∇T)+Q
where *ρ*, *k*, *Cp*, and *Q* are the temperature-dependent density, heat conductivity, specific heat, and volumetric heat input, respectively.

The powders were fused by a CO_2_ laser with a spot diameter of 0.6 mm and wavelength of 10.55–10.63 μm. The repeated reflections and absorption of laser energy occur due to the porosity of the powder bed. Additionally, the Beer-Lambert attenuation law [15] was adopted to represent the laser power variation in the depth direction.
(2)$$Q\left(x,y,z\right)=Q_{0}\left(x,y\right)\cdot \beta \cdot \exp\left(-\beta z\right)$$
where *Q*_0_ represents the surface thermal distribution. *β* is the extinction coefficient and set to 7500 [15]. The term *exp*(−*βz*) indicates the attenuation of the laser power with the depth from the powder surface. *Q*_0_ is expressed by the Gaussian equation [16] as
(3)$$Q_{0}\left(x,y\right)=2AP/\pi r_{laser}^{2}\cdot \exp\left(-2\left(x^{2}+y^{2}\right)/r_{laser}^{2}\right)$$
where *A* is the power absorption coefficient and set to 0.9 [17], *P* is the laser power, and *r_laser_* is the laser spot radius. Then, the volumetric heat input is expressed as
(4)$$Q\left(x,y,z\right)=2AP\beta /\pi r_{laser}^{2}\cdot \exp\left(-2\left(x^{2}+y^{2}\right)/r_{laser}^{2}\right)\cdot \exp\left(-\beta z\right)$$

To minimize the melting energy required from laser, the initial temperature condition is set to
(5)$$T\left(x,y,z,0\right)=T_{b}=170$$

The absorbed laser energy is much greater than the radiant energy owing to the rapid process of melting–consolidation [18]. Thus, the boundary condition on the external surfaces is described as
(6)$$-nk\Delta T=q-\gamma \left(T-T_{b}\right)$$
where *n* is the unit normal vector, *γ* is the natural convection coefficient, and *q* is the thermal input. The process conditions and laser constants are listed in Table 1.

Dimensional inaccuracy is adversely influenced by high cooling rates, which results in critical thermal and recrystallization-induced stress. The fused amorphous phase starts to form semi-crystalline structures during the cooling step. The time evolution of the crystallinity can be predicted by [19]
(7)$$\partial \lambda /\partial t=NK\left(T\right)\left(1-\lambda \right){\left[\ln\left(1/\left(1-\lambda \right)\right)\right]}^{\left(N-1\right)/N}$$
where *λ* is the degree of crystallinity, *N* is the Avrami exponent, which expresses the propensity of the material to nucleate and grow crystalline phase. Here, *N* is 6.8, which is a typical value for PA12 [20]. *K(T)* is the non-isothermal rate of crystallization kinetics, which is given by
(8)$$K\left(T\right)=\ln{\left(2\right)}^{1/N}\left(1/t_{1/2}\right)$$
where *t*_1/2_ is the time taken to crystallize half amount of material and depends on the cooling rate. The computing method of *t*_1/2_ is described in Appendix A. To demonstrate the effective thermal properties accurately in the cooling step, the following mixing laws are adopted.
(9)$$\rho_{c}\left(\lambda ,T\right)=\lambda \rho_{s}\left(T\right)+\left(1-\lambda \right)\rho_{m}\left(T\right)$$
(10)$$Cp_{c}\left(\lambda ,T\right)=\lambda Cp_{s}\left(T\right)+\left(1-\lambda \right)Cp_{m}\left(T\right)$$
(11)$$k_{c}\left(\lambda ,T\right)=\lambda k_{s}\left(T\right)+\left(1-\lambda \right)k_{m}\left(T\right)$$
(12)$${\alpha_{T}}_{c}\left(\lambda ,T\right)=\lambda {\alpha_{T}}_{s}\left(T\right)+\left(1-\lambda \right){\alpha_{T}}_{m}\left(T\right)$$
where *ρ**_c_**(λ,T)*, *Cp_c_(λ,T)*, *k_c_(λ,T)* and *α**_Tc_**(λ,T)* are the effective density, specific heat, thermal conductivity, and thermal expansion in the cooling step, respectively. The crystalline state and amorphous state of the bulk polymer material are represented by the subscripts s and m, respectively.

### 2.2. Thermo-Mechanical Model

#### 2.2.1. Shrinkage and Warping Model

For SLS process (Figure 1), the semi-crystalline polymer powders experience fusion, solidification, and recrystallization in the heating–cooling stage.

In Figure 1b, the powder fusion-induced shrinkage is ubiquitous due to material state transferring from powder to solid state. Additionally, the shrinkage is associated with the thermal contraction of polymer upon cooling. Furthermore, the phase transition from amorphous to crystalline phase also causes the volumetric shrinkage due to the closely packed crystal structure. As a result, the total shrinkage is described by
(13)$$\delta_{total}=\delta_{re}+\delta_{powder}+\delta_{T}$$
where *δ_re_* is the shrinkage caused by polymer recrystallization, *δ_powder_* is the shrinkage resulting from the powder block and particle characteristics at bed temperature. The density change of powders upon cooling is temperature-dependent, which was measured in our previous study [21]. *δ_T_* is the thermal contraction of polymer upon temperature drop. Thereafter, warping is mainly influenced by the thermal elasto-plastic deformation and volumetric shrinkage.

For the semi-crystalline polymer, considering recrystallization-induced strain, the total strain increment is expressed as
(14)$$d\varepsilon_{total}=d\varepsilon_{e}+d\varepsilon_{p}+d\varepsilon_{th}+d\varepsilon_{re}$$
where *d**ε_total_*** is total strain increment. *d**ε_e_***, *d**ε_th_***, *d**ε_p_***, and *d**ε_re_*** are the elastic strain increment, thermal strain increment, plastic strain increment, and recrystallization-induced strain increment, respectively. The stress increment *d**σ*** is given by
(15)$$d\sigma =D\left(d\varepsilon_{total}-d\varepsilon_{p}-d\varepsilon_{th}-d\varepsilon_{re}\right)$$
where ***D*** represents the elastic matrix, and the plastic strain increment *d**ε_p_*** is calculated by
(16)$$d\varepsilon_{p}=d\tau \cdot \left(\partial f/\partial \sigma \right)$$
where ***τ*** is the hardening coefficient, ***f*** is the yield function and expressed as
(17)$$f=\sigma_{von}-\sigma_{yield}$$
where ***σ_von_*** is the von Mises stress and ***σ_yield_*** is the yield stress. The thermal strain increment *d**ε_th_*** is given by
(18)$$d\varepsilon_{th}=\alpha_{T}\cdot \Delta T$$
where *α_T_* is the thermal expansion coefficient and *∆T* is the temperature difference. Here, the temperature-induced shrinkage *δ_T_* is mainly represented by *d**ε_th_***.

The expression of recrystallization-induced strain increment *d**ε_re_*** is given by
(19)$$d\varepsilon_{re}=\varepsilon_{\Delta l}\cdot \Delta \lambda \left(T\right)$$
where ***ε_∆l_*** is the recrystallization-induced strain corresponding to the full crystallization of semi-crystalline polymer at a specific cooling rate, and *λ(T)* is the temperature-dependent degree of crystallinity.

Recrystallization occurs in semi-crystalline polymers during the cooling process, which has an obvious impact on the crystallite size. Thus, the change rate of temperature-dependent crystallite size in the cooling stage is introduced to evaluate the full recrystallization-induced strain ***ε_∆l_***.
(20)$$\varepsilon_{\Delta l}=\left(l_{2}-l_{1}\right)/l_{1}$$
where *l_1_* and *l_2_* are the crystallite sizes at initial crystallization temperature and room temperature in the cooling process. Equation (19) can be expressed as the incremental form.
(21)$$\Delta \varepsilon_{re}=\varepsilon_{\Delta l}\cdot \left(\lambda \left(T_{1}+\Delta T\right)-\lambda \left(T_{1}\right)\right)$$

Then divide both sides of Equation (21) by *∆T*.
(22)$$\Delta \varepsilon_{re}/\Delta T=\varepsilon_{\Delta l}\cdot \left(\lambda \left(T_{1}+\Delta T\right)-\lambda \left(T_{1}\right)\right)/\Delta T=\varepsilon_{\Delta l}\cdot \lambda^{\prime }\left(T\right)$$
where *λ′(T)* is the derivative of the degree of crystallinity with respect to specific temperature. Then
(23)$$d\varepsilon_{re}=\varepsilon_{\Delta l}\cdot \lambda^{\prime }\left(T\right)\cdot \Delta T$$

It is worth noting that the recrystallization-induced shrinkage *δ_re_* is mainly considered by *d**ε_re_***. Substituting Equation (23) into Equation (18) yields
(24)$$d\varepsilon_{th}+d\varepsilon_{re}=\left(\alpha_{T}+\varepsilon_{\Delta l}\cdot \lambda^{\prime }\left(T\right)\right)\cdot \Delta T$$

#### 2.2.2. FEM Model Setup

The thermo-mechanical model was implemented through the Abaqus^®^ with its corresponding subroutine program. The model was divided into a substrate and slice layers of the specimen. The substrate was fixed, and the constraint of the substrate on the specimen was realized by direct contact, which was the same way of constraints between adjacent layers. For the thermal boundary conditions, the convective heat-transfer applied on the side and top surface of the powder layer was treated identically. The spreading process of each layer was implemented by the birth-death element method, and the elements of the layer were activated at a predefined analysis step. Meanwhile, a recoated layer was able to cool down for 0.2 s at this analysis step. The thermal boundary conditions can be applied on a sequentially varying surface which represents the present shape of the printing part.

The temperature-dependent material properties were imported into the numerical model by the inherent material module, which contains properties such as thermal expansion coefficient, density, heat conductivity, specific heat, and mechanical properties of elastic modulus and yield stress. A field variable, *ψ*, was defined to represent the “powder–liquid–solid” phase transition according to the temperature.
(25)$$\psi =\left\{ \begin{array}{ll} 0 & \left({T\;<\;T}_{m},\;\mathrm{powder}\right)\\ 1 & \left({T\;>\;T}_{m},\;\mathrm{liquid}\right)\\ 2 & \left({T\;<\;T}_{m},\;\mathrm{bulk}\right) \end{array}\right.$$
where *T_m_* is the melting point. Since the temperature regions of the powder phase and bulk phase overlap, an additional judgment according to whether the temperature has ever exceeded the melting point (*ψ* > 0) is proposed. Additionally, the degree of crystallinity and crystallite size are used to calculate the recrystallization-induced strain *d**ε_re_***, and the expression form of *d**ε_re_*** is similar to the thermal strain increment *d**ε_th_*** described in Equation (18). Therefore, the recrystallization-induced strain *d**ε_re_*** can be realized by changing the thermal expansion coefficient in the simulation based on Equation (24).

The heat-transfer analysis, including specific heat source equation was implemented through the DFLUX subroutine program. During the cooling stage, by introducing USDFLD subroutine program, a specific variable was defined to represent the temperature-dependent degree of crystallinity, and the effective properties described in Equations (9) to (12) were considered through the field variable. Therein, the polymer materials are in the amorphous state at the temperature above the melting point. Thus, the thermal property slightly below the onset of melting range was adopted as the one in the amorphous state. Detailed material evaluations are given in Section 3.1.

### 2.3. Material Characterization and Experimental Design

#### 2.3.1. Material Characterization

The specific properties of semi-crystalline PA12 provided by the material datasheet from supplier (FS3300PA from Farsoon Technologies, Changsha, China) are listed in Table 2.

The thermal expansion coefficient was detected using a thermal expansion instrument (NETZSCH, Bavaria, Germany, 20095590). The test sample is a cylinder with diameter of 6 mm and length of 25 mm. The testing samples were heated up from room temperature to 250 °C with a heating rate of 10 °C/min. Then the cooling step was carried out with a natural cooling rate.

The specific volume of the polymer performs as a function of temperature during the heating and cooling process [5]. The temperature-dependent density was evaluated by the reciprocal of specific volume and measured in our previous study [21]. Additionally, the porosity of the powder bed was 0.4 [22].

The thermal conductivity was measured through the DRPL-II thermal conductivity instrument (Xiangtan instrument co. LTD, 8770, Xiangtan, China). The detected sample was designed as a cube 2 × 2 × 0.3 cm^3^ and placed at the position between the baseboard and heating panel. The thermal conductivity was recorded automatically until the thermal equilibrium was reached. Additionally, the thermal conductivity of liquid and powder phase were disposed of linear relationship [23].

The specific heat was measured through differential scanning calorimetry (DSC) equipment (NETZ5CH, DSC 200 F3). The PA12 powders of 7.8 mg were placed in the sample holder and heated up with 10 °C/min from room temperature to 250 °C. After five minutes, it was cooled at 10 °C/min to 20 °C.

The elastic modulus and plastic stress were measured according to the Type 1 specimen of GB/T 16421-1996 (ISO 527-2:1993) with a loading speed of 1 mm/min. To acquire the degree of crystallinity in different cooling rates, the PA12 powders were heated up with the rate of 10 °C/min from room temperature to 250 °C and held for five minutes. Sequentially, the cooling operation was conducted with different cooling rates in the DSC experiments. The crystallization enthalpy *∆H* at different cooling rates was obtained. Additionally, the theoretical enthalpy of full crystallization *H_100_* was taken as 134 J/g [24]. Thus, the maximum degree of crystallinity *λ_max_* at a specific cooling rate is expressed as
(26)$$\lambda_{\max}=\Delta H/H_{100}\times 100\%$$

The X-ray diffractometer (XRD from Bruker D8 Advance, Billerica, MA, USA) was employed to measure the crystallite structure. The PA12 powders were heated to 220 °C at the rate of 10 °C/min and held for five minutes, which is followed by the natural cooling step. The crystallite structure was measured at 180 °C, 142 °C, 103 °C, 64 °C, and 25 °C within the cooling stage. The crystallite size of PA12 is calculated based on Scherrer-method. Scherrer formula is expressed as
(27)$$D_{L}=F\omega /B\cos\zeta$$
where *F* is the Scherrer constant 0.89, *D_L_* is the average crystallite size perpendicular to the crystal surface, and *ω* is the wavelength of incident X-Ray. *B* (in radians) is the width of the diffraction peak at the half-height position. *ζ* (in degrees) is the Bragg diffraction angle.

#### 2.3.2. Orthogonal Experiment

The cubic specimen (50 × 10 × 1 mm^3^) was employed to measure the shrinkage and warping. The controlled parameters were laser power, layer thickness, scanning speed, hatching space, and “layer-layer angle” (shown in Figure 2). It is worth noting that the dimensional accuracy of the as-printed components is greatly affected by the scanning strategy and algorithm [25]. In this work, the controlled parameter “layer-layer angle”, showing the difference between scanning directions of adjacent layers, is a reflection of scanning strategy of simple raster-scanning. Different angles of θ result in varied thermal gradients upon scanning. Thereby the temperature, residual stress, and entire shrinkage/warping are influenced.

Different levels of each parameter were given in Table 3. *L*_16_ (4^5^) orthogonal array [26] was used as listed in Appendix A.

In this experiment, the EP-P3850 machine (E-Plus-3D Company) was used for SLS process. Before printing, the laser window mirror and temperature sensor were wiped carefully with alcohol, and the surface of the powder bed should be flat enough by several times of rolling. To verify the consistency of different positions of the powder bed, 16 duplicates with the arbitrarily chosen parameters *P*_2_*v*_1_*h*_2_*t*_3_*θ*_4_ were placed uniformly and accurately on the position shown in Figure 3a, and the measured length values shown in Figure 3b exhibited a good consistency.

During printing, 150 layers were recoated in the preheating stage to make the working environment stable enough and fully meet the powder bed temperature of 170 °C. The chamber temperature was 120 °C. After printing, the heating devices were stopped, and the environmental temperature was noted hourly. The process temperature over the whole SLS process is illustrated in Figure 4.

Each parameter group included three replicas (Figure 5a). To measure the length *L* and warping *u* accurately, the cross-section was taken using a carefully aligned digital camera. The averages of length and warping value were calculated as objective values to compare with the modeling results. Additionally, the shrinkage of specimens was given as
(28)$$L_{\varepsilon }^{i}=\left(50-L_{i}\right)/50\times 100\%$$
where *L^i^_ε_* is the shrinkage, *L_i_* denotes the length of the *i*-th specimen.

The analysis of variance (ANOVA) was carried out for process optimization to avoid shrinkage or warping, and the influence order of different parameters on the objective value is also investigated. It was noteworthy that the source of error in ANOVA was varied with the amount of repeated experiments due to the inexistence of the empty column in the orthogonal table [27]. To acquire the quantitative correlations between shrinkage/warping and the parameters, the stepwise data fitting method of quadratic polynomial and sequential analysis based on orthogonal results were implemented via the professional data analysis software DPS [28]. The multiple quadratic regression equation can be expressed as
(29)$$y=q_{0}+\sum_{i=1}^{n}q_{i}x_{i}+\sum_{i=1}^{n}q_{ii}x_{ii}^{2}+\sum_{i<j}\sum q_{ij}x_{i}x_{j},$$
where $q_{0}$, $q_{i}$, and $q_{ii}$ are the coefficient of the constant term, one-degree term, and quadratic term, respectively. $q_{ij}$ is the coefficient of the interaction term.

## 3. Results and Discussion

This section introduces the capabilities of the thermo-mechanical model in predicting the temperature distribution, residual stress profile, and shrinkage/warping of laser-sintered specimens. Therein, the initial operation window of energy density is determined by the prediction of temperature field and melting pool size. The residual stress distribution and the influence of cooling rate on the shrinkage and warping are predicted. The shrinkage/warping results of orthogonal experiment and simulation are compared systematically to validate the reliability of the numerical model. The optimal set of parameters and the quantitative relationships of shrinkage/warping are acquired.

### 3.1. Material Evaluation

The temperature-dependent thermo-mechanical properties, degree of crystallinity, and crystallite size of PA12 are established by proper experimental evaluation. The thermal properties of PA12 including thermal expansion *α_T_*(*T*), specific heat *Cp*(*T*), density *ρ*(*T*), and heat conductivity *k*(*T*) are demonstrated in Figure 6. The mechanical properties of elastic modulus *E*(*T*) and yield stress *σ_yield_*(*T*) are plotted in Figure 7.

In Figure 6a, the thermal expansion increases with the increase of temperature, and the state transition from solid to liquid has a slight influence on the change of temperature-dependent thermal expansion. In Figure 6b, the density difference in sintered solid part and powders is resulted from powder-fusion, and a notable change on the curve is observed at the onset and offset of the melting stage (*T_ms_* and *T_mf_* represent the onset and offset of melting, respectively). Similarly, the change occurs at the location of the solid-liquid phase transition in the temperature-dependent heat conductivity shown in Figure 6c. In Figure 6d, the specific heat shows a sharp melting peak, and is linearly proportional to the temperature in the liquid or powder phase [21].

In Figure 7a, the elastic modulus decreases gently with the increase of temperature until the glass transition temperature of 51 °C is reached [19]. Thereafter, the elastic modulus declines rapidly until the melting point of 183 °C. The change rate of elastic modulus with temperature reveals the various states of the polymer such as glass state and viscous flow state. In Figure 7b, the yield stress decreases obviously with the increase of temperature and achieves a relatively small value at the liquid phase.

The crystallization enthalpy *∆H* at different cooling rates was obtained by the DSC measurements. The maximum degree of crystallinity *λ_max_* at a specific cooling rate is acquired and given in Table 4.

The curves of crystallization kinetics (Figure 8b) were derived from DSC experiments and the principle of crystallization kinetics. It shows that the curves of crystallization kinetics in different cooling rates show a similar trend, and the degree of crystallinity increases with the increase of cooling rate. When the cooling rate is redcued, the crystallization is saturated at a relatively high temperature. Meanwhile, the shrinkage increases with the increase of crystallinity, which is the main reason of the recrystallization-induced shrinkage. The molecular chain arrangement greatly compacts with the increase of crystallinity. Additionally, the natural cooling rate is approximately equivalent to 0.8 °C/min in the cooling stage of SLS.

Crystallite size usually influences the recrystallization-induced strain. The XRD spectrogram of PA12 shows the phase absorption peaks corresponding to different temperatures upon cooling (Figure 9a), and the fitted temperature-dependent crystallite size of PA12 is exhibited in Figure 9b. PA12 shows different polymorphs of α, α′, and γ [29]. Each polymorph has a distinct response to the temperature change. The as-exhibited crystallite size is obtained by averaging the crystalline size of each polymorph.

As described in Equations (20) and (23), the change of crystallite size induces the recrystallization-induced strain *ε_re_*, then causes the recrystallization-induced shrinkage *δ_re_*. In Figure 9b, as the temperature decreases from 180 °C to 120 °C, the change of crystallite size is relatively gentle. An obvious decline occurs from 120 °C to 55 °C. Thereafter, the crystallite size increases from 55 °C to room temperature. It should be noted that the absolute change of crystallite size from 180 °C to room temperature is around 50 nm (∆*l* = *l*_2_−*l*_1_), which is used for the numerical modeling.

The temperature-dependent thermo-mechanical properties, degree of crystallinity, and crystallite size are experimentally measured for further implementation in the proposed numerical modeling. The temperature, residual stress, and shrinkage/warping of PA12 in SLS are referring to these specific material properties in the following sections.

### 3.2. Thermal Distribution and Effective Energy Density Window

The energy input per unit volume (energy density) *E_vol_* from the laser are determined by [21]
(30)$$E_{vol}=P/vht$$
where *P* is the laser power, *v* is the scanning speed, *h* is the hatching space, and *t* is the layer thickness, respectively. Micro-scale melting pool is predicted by the heat-transfer model with the integration of proper laser source and temperature-dependent material functions. Three constraint conditions are taken into account for the selection of an appropriate region of energy density, which can ensure the fully sintering of powder bed with high efficiency [30].

The constraint conditions are described as
(1)*T_max_* ∊ stable sintering range (SSR);(2)width > h;(3)t < depth < 2t.
where SSR is the optimal temperature range to secure the successful sintering. It starts from the offset of melting temperature *T_mf_* to the onset of decomposition temperature *T_ds_* [21]. The melting pool corresponds to the region with the temperature higher than the melting point, and the melting pool temperature should be within the SSR. The width and depth of the melting pool (shown in Figure 10) are the key values to determine the energy density. Additionally, the decomposition behavior of PA12 was observed in the TGA diagram [21], which showed a wide range of SSR from 195 °C to 370 °C.

A series of process parameters are listed in Appendix A. *T_max_*, width, and depth of the melting pool are shown in Figure 11. In Figure 11a, it is shown that the width and depth of the melting pool increase with the increase of laser power. The volume energy density and the laser power exhibit the same variation trend when other parameters are identical [21]. According to the constraint conditions of (2) and (3), the feasible region of energy density is identified of 0.085–0.263 J/mm^3^. In Figure 11b, the depth and width increase with the decrease of scanning speed. In Figure 11c, the depth grows with the increase of hatching space and energy density. The width is not observed here, as it changes with constraint conditions (2). In Figure 11b,c, according to the constraint conditions of (2) and (3), both of their upper limits of energy density are around 0.26 J/mm^3^, which is consistent with Figure 11a and serves as a validation. Figure 11d corresponds to the constraint conditions (1), the lower limit of energy is around 0.097 J/mm^3^. In general, the range of SSR can be regarded as a symbol of material processability, and appropriate energy density for PA12 is calculated as 0.097–0.263 J/mm^3^.

The determination of the effective energy density window can narrow down the parameter selection scope, thereby improving the efficiency of the parameter optimization process. The effective energy density is determined based on the temperature distribution of the melting pool. Subsequently, residual stress including elasto-plastic stress, thermal stress, and recrystallization-induced stress is calculated through the thermo-mechanical model.

### 3.3. Numerically Predicted Residual Stress, Shrinkage and Warping

The residual stress includes elasto-plastic stress, thermal stress, and recrystallization-induced stress. The total residual stress of the cooled specimen is shown in Figure 12 with the combination of parameters *P*_1_*v*_4_*h*_3_*t*_4_*θ*_4_. It appears that both *S_x_* and *S_y_* are symmetrically distributed, and *S_z_* exhibits uniformity overall (length, width, and height of specimen represent the *x*, *y*, and *z* direction, respectively), which may be a result of the symmetry of the specimen geometry, its particularly thin thickness, and the natural cooling condition. Residual stress should be monitored continuously during the laser-sintering process. When the parameters are not adjusted appropriately during the printing process, the insufficient sintering or overheating can lead to porosity or defects, induce the variation of residual stress, and further adversely impact the performance of product-scale components.

The shrinkage and warping of identical specimens were assessed with different cooling rates at 0.8 °C/min, 5 °C/min, 10 °C/min, 15 °C/min, 20 °C/min, and 25 °C/min to investigate the influence of recrystallization phenomenon. In Figure 13, with the increase of cooling rate from 0.8 °C/min to 25 °C/min, the shrinkage increases from 2.7% to 3.2% and the warping increases slightly from around 0.18 mm to 0.22 mm. As expected, the shrinkage and warping both show a positive correlation with the cooling rate of semi-crystalline polymer, due to the fact that the crystallinity grows as the increase of cooling rate. The polymer materials consist of a large quantity of molecular chains. The compaction of molecular chains induced by recrystallization causes the recrystallization-induced strain, thereby leading to an increase in shrinkage and warping.

### 3.4. Experimental Validation of Shrinkage and Warping

An orthogonal experimental design was adopted for multi-factor analysis. The shrinkage and warping were the objective values, and the ANOVA was performed to obtain the optimal combination of parameters. The stepwise data fitting method using quadratic polynomial was accepted to derivate the quantitative relations between parameters and shrinkage/warping.

Figure 14a,b shows the length and warping of specimens in the experiment and in modeling (detailed data shown in Appendix A), respectively. In Figure 14a, the length curves of the experiment and modeling exhibit a similar trend. This indicates that the model is referential in the parameter selection. The length obtained by numerical prediction is generally larger than that measured by the experimental method, with error in the range from 0.23% to 1.32%. This is mainly due to the deviation of the cooling condition. The cooling rate in the experiment is probably larger than 0.8 °C/min. Meanwhile, the shrinkage values from the experiment and modeling are in the range of 2.5% to 5%, which is also in agreement with the experimental measurement of [5]. By contrast, the warping curves (Figure 14b) show a relatively close trend and magnitude with the existence of individual differences. For groups 4 and 9, the warping values of experiment are greater than these of simulation. This may be due to the fact that the largest layer thicknesses are selected for groups 4 and 9, and thus insufficient fusion occurs and causes delamination between layers. Such delamination can be further enlarged by post-processing via sandblasting. For other cases with sufficient fusion among layers, the error induced by post-process is not significantly observed from experimental measurements. However, a careful blasting process is also suggested with proper type of sand and air pressure. Consequently, the comparison of experiment and modeling provides a fundamental validation for the reliability and feasibility of the thermo-mechanical model.

Table 5 shows the optimal combination of parameters acquired from the columnar graph (Figure 15). The maximum difference of R reflects the significance of each parameter. Meanwhile, the ANOVA was performed to quantify the significance of each parameter intuitively. The ANOVA results of shrinkage and warping are listed in Appendix A, respectively. The *F* value indicates the importance of each parameter on the objective values. It appears that all the parameters studied are significant and the influence orders are identical to those in Table 5.

As listed in Table 5, the shrinkage values of the experiment and the modeling both exhibit the significance of process parameters following the order of *h* > *t* > *P* > *θ* > *v*. *v* and *θ* are the factors showing a different influence on the shrinkage. For warping, only *P* is different in terms of the optimal set of parameters of experiment and modeling. The parameter *t* changes obviously in the parameter influence order, which indicates that *t* deserves more attention in investigating the warping. Additionally, the significance of *θ* is greater than the scanning speed *v*. Although *θ* is not considered in the conventional evaluation of laser energy density. This indicates that scanning strategy results in the dimensional inaccuracy and parameter optimization in some way. Furthermore, the experimental verification of optimal combinations of parameters *P*_2_*v*_4_*h*_4_*t*_4_*θ*_2_ and *P*_1_*v*_4_*h*_4_*t*_4_*θ*_2_ in Table 5 of visualized warping were carried out and shown in Figure 16, which exhibits a satisfactory warping result. Additionally, the optimal sets of parameters of shrinkage and warping are obviously different. While the shrinkage is primarily affected by the recrystallization, powder densification, and thermal retraction, the warping is nevertheless mainly influenced by the thermal elasto-plastic deformation.

The stepwise data fitting method based on quadratic polynomials was employed to obtain the following empirical formulas based on the shrinkage and warping of modeling predictions (parameter *θ* is used as radian). The optimal combinations of parameters derived from the corresponding quantitative relationships (Equations (31) and (32)) are listed in Table 6.
(31)$$L_{\varepsilon }=8.7668+4.1563\times 10^{-1}P+1.5203\times 10^{2}h+2.1254\times 10^{1}\theta \;\;+4.4139\times 10^{-4}P^{2}+4.8573\times 10^{-7}v^{2}+2.4731\times 10^{2}h^{2}-8.0190\times 10^{1}t^{2}\;\;-1.0401\theta^{2}-8.4610\times 10^{-5}Pv-2.0602\times 10^{-1}P\theta +5.7204\times 10^{-4}vh\;\;-2.0510\times 10^{-3}v\theta +1.8570\times 10^{2}ht-3.9823\times 10^{1}t\theta$$
(32)$$u=-0.3071+5.6709\times 10^{-2}P+1.0379\times 10^{1}h-1.3005\times 10^{1}t\;\;+0.2331\theta -2.0151\times 10^{-3}P^{2}+4.1545\times 10^{-8}v^{2}+0.3508\theta^{2}\;\;+0.2985Ph-2.7819\times 10^{-2}P\theta -4.3246\times 10^{-3}vh+4.5817\times 10^{-3}vt\;-2.8214\times 10^{1}ht+0.5473h\theta -1.2429t\theta$$

In Table 6, it appears that the formulas of shrinkage and warping exhibit acceptable accuracy and reliability. The parameter combinations with the minimum of shrinkage and warping are not integers, which cannot be deduced just from the analysis of orthogonal results. However, the quantitative relationships can provide a comprehensive guide for identifying the influence of each process parameter on the product performances via shrinkage and warping. Meanwhile, the optimal combination of process parameters is also achievable with respect to the desired effects.

### 3.5. Shrinkage and Warping of Complex Structure

To validate the application of the proposed model in part-scale, a representative design was employed for macro-scale shrinkage and warping prediction (Figure 17). The shrinkage was represented by the change of the longest edge of the structure, and the warping is shown in Figure 17c.

Cases with different powers, including 20 W, 25 W, 30 W, and 35 W, were applied to the part-scale numerical evaluation, and other parameters including scanning speed, layer thickness, hatching space, and cooling rate were kept fixed as 4500 mm/s, 0.1 mm, 0.3 mm, and 0.8 °C/min, respectively. The values of mises stress *S_mises_*, horizontal displacement *u_1_* and vertical displacements *u_3_* associated with shrinkage and warping are obtained and demonstrated in Figure 18. The power-dependent shrinkage and warping are shown in Figure 19.

Figure 18 shows that the *S_mises_* increases slightly with the increase of power, the stress distributions at different powers are almost identical and are symmetrical. An abrupt change occurred at the corners of the structure, specifically the corners of the long side on the top layer. One reason causing this abrupt change is the geometric construction; another one is the nonuniformity of the temperature field. The temperature distribution differs little with the increase of power, since the temperature distribution is mainly influenced by post-processing parameters such as cooling rate and heat preservation time. The relatively low cooling rate and extended heat preservation are associate with the uniform of temperature field. In Figure 19, with the increase of power from 20 W to 35 W, the shrinkage increases from around 2.34% to 2.60%, and the warping increases from around 0.16 mm to 0.21 mm. This is induced by the increase of thermal strain, as well as the increase of energy density.

Therefore, the desired shrinkage and warping are correlated with multiple factors and can be guided by the thermo-mechanical model with the systematic parametric design method, which has been conducted and experimentally validated. Therefore, the numerical model from micro- to macro-scale is generally applicable for predicting the shrinkage and warping of entire parts via SLS.

## 4. Conclusions

In this study, a numerical model system with a combination of heat-transfer, thermo-mechanical, and material phase transfer was developed to predict temperature profile and residual stress, so as to obtain the shrinkage and warping at part-scale via SLS. The model firstly predicts the micro-scale melting pool, which identifies the feasible operation window of laser energy. Then, by incorporating this with the thermo-mechanical model, the stress distribution, the part-scale shrinkage, and warping can be predicted systematically. It is worth noting that the powders are considered to be continuum media with homogenized thermal and mechanical properties in the macro-scale thermo-mechanical model. Meanwhile, the model accounts for phase transformations (physical state changes) from raw material via melting/solidification followed by recrystallization in the cooling stage. The shrinkage model including thermal retraction, recrystallization, and powder densification is universal for semi-crystalline polymers. Thus, the model is universal for temperature distribution and residual stress prediction regardless of the type of materials, especially the semi-crystalline polymers in the SLS process. The experimental method via orthogonal approach is implemented in order to validate the shrinkage and warping adequately, and the significance order of influencing parameters is also investigated to provide a guidance for dimensional accuracy control. The quantitative correlations between shrinkage/warping and the process parameters are established for future parameter optimization to avoid dimensional inaccuracy. Finally, a complex geometry is used for macro-scale shrinkage and warping prediction based on the proposed thermo-mechanical system to demonstrate its applicability.

## Figures and Tables

**Figure 1 polymers-12-01373-f001:**
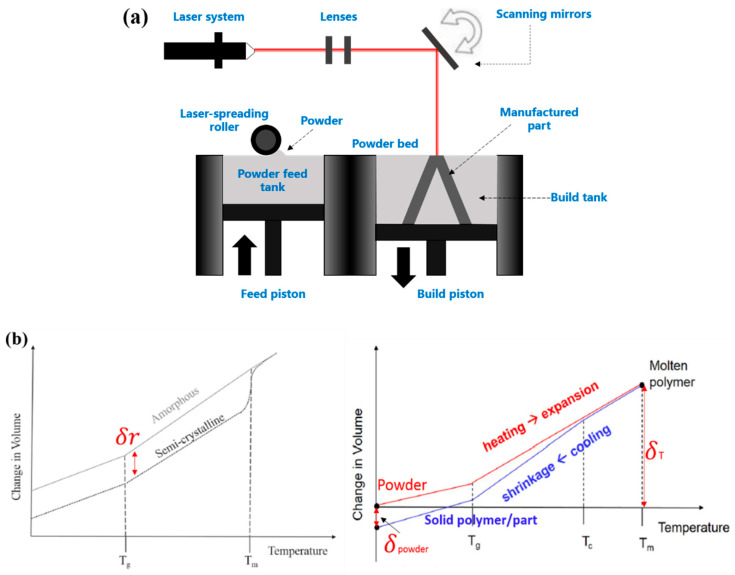
(**a**) Schematic of laser sintering principle; and (**b**) illustration of critical shrinkage factors upon cooling stage in the SLS process.

**Figure 2 polymers-12-01373-f002:**
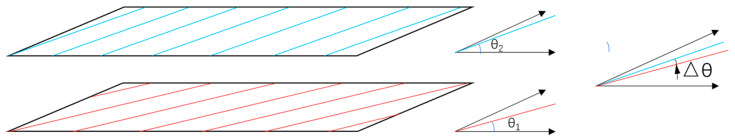
The diagrammatic sketch of layer-layer angle (the angle difference between adjacent layers, ∆*θ* = *θ*_1_−*θ*_2_).

**Figure 3 polymers-12-01373-f003:**
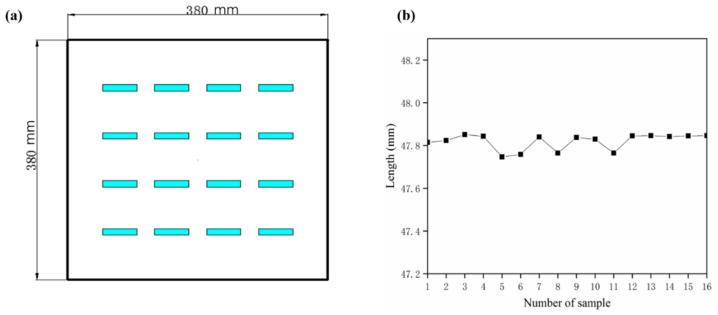
The diagrammatic sketch of location consistency verification: (**a**) locations; (**b**) length values.

**Figure 4 polymers-12-01373-f004:**
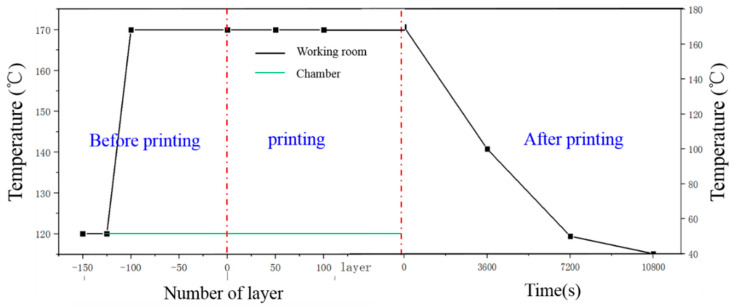
Temperature condition during the SLS process.

**Figure 5 polymers-12-01373-f005:**
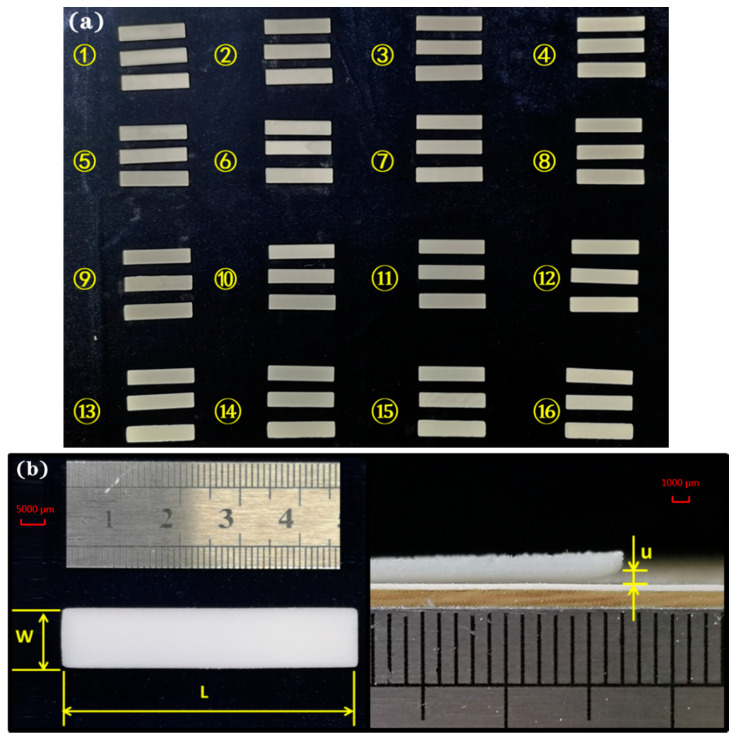
Cubic specimens and measurement approach: (**a**) the measured specimens; (**b**) the dimensions for shrinkage and warping evaluation.

**Figure 6 polymers-12-01373-f006:**
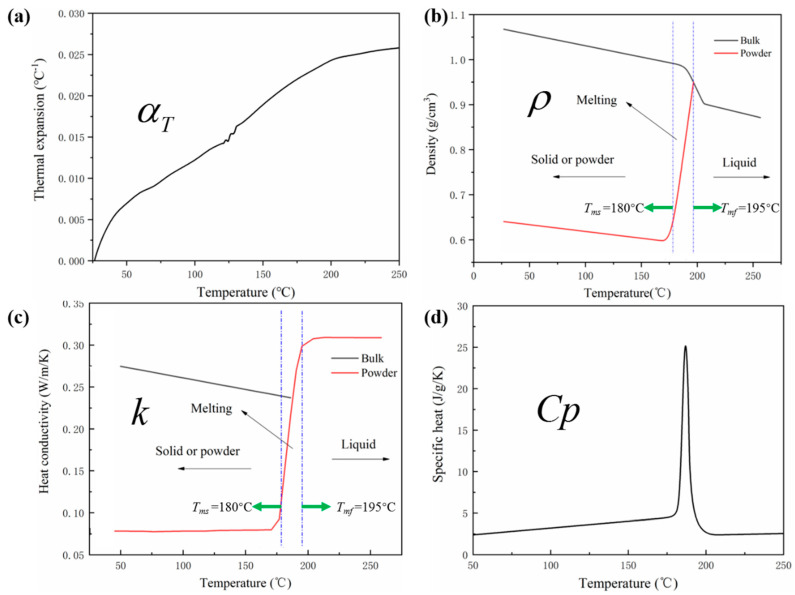
Thermal property of PA12: (**a**) thermal expansion *α_T_*(*T*); (**b**) density *ρ*(*T*); (**c**) heat conductivity *k*(*T*); (**d**) specific heat *Cp*(*T*).

**Figure 7 polymers-12-01373-f007:**
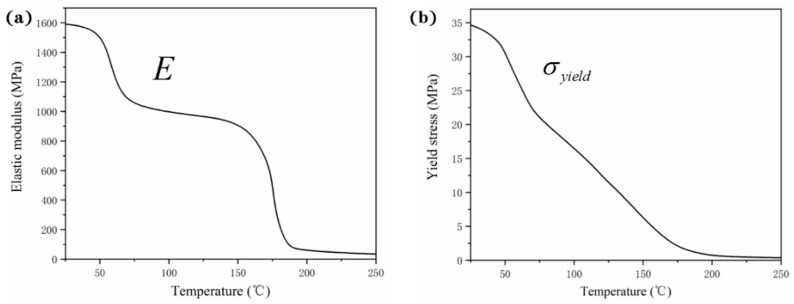
Temperature-dependent (**a**) elastic modulus *E*(*T*) and (**b**) yield stress *σ_yield_* (*T*) of PA12.

**Figure 8 polymers-12-01373-f008:**
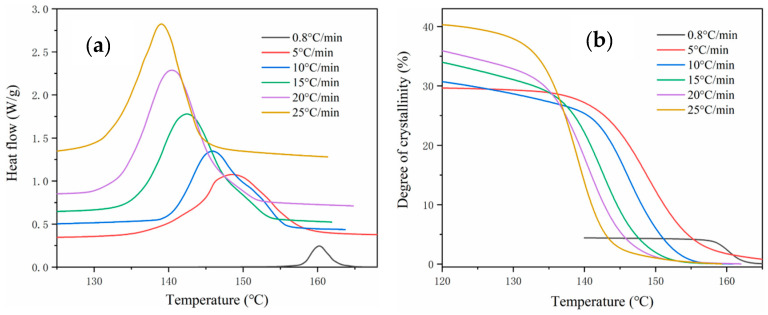
DSC diagram of PA12: (**a**) heat flow; (**b**) degree of crystallinity.

**Figure 9 polymers-12-01373-f009:**
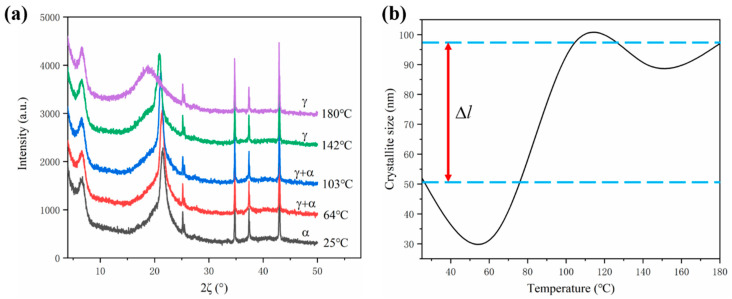
The XRD results in the cooling stage: (**a**) XRD spectrogram; (**b**) crystallite size.

**Figure 10 polymers-12-01373-f010:**
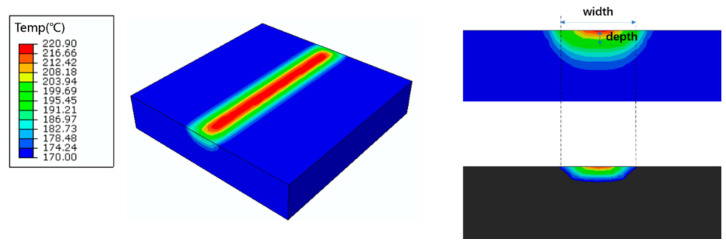
Schematic illustration of the laser sintering process and melting pool.

**Figure 11 polymers-12-01373-f011:**
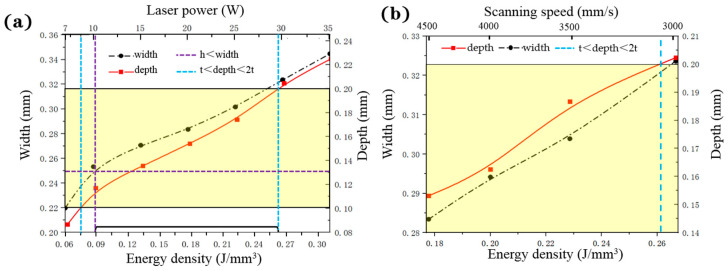
The effective width, depth and *T_max_* of melting pool: (**a**) effective width and depth with respect to the varied laser powder and corresponding energy density (NO. 1–7 in Appendix A); (**b**) effective width and depth with respect to the varied scanning speed and corresponding energy density (NO. 8–11 in Appendix A); (**c**) effective depth with respect to the varied hatching space and corresponding energy density (NO. 12–15 in Appendix A); (**d**) *T_max_* with respect to the corresponding energy density.

**Figure 12 polymers-12-01373-f012:**
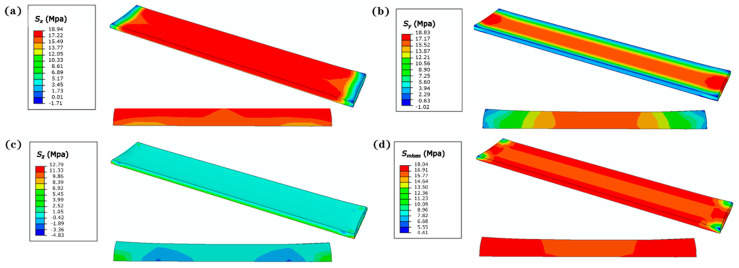
Residual stress distribution of cubic specimens: (**a**) *S_x_*; (**b**) *S_y_*; (**c**) *S_z_*; (**d**) *S_mises_*.

**Figure 13 polymers-12-01373-f013:**
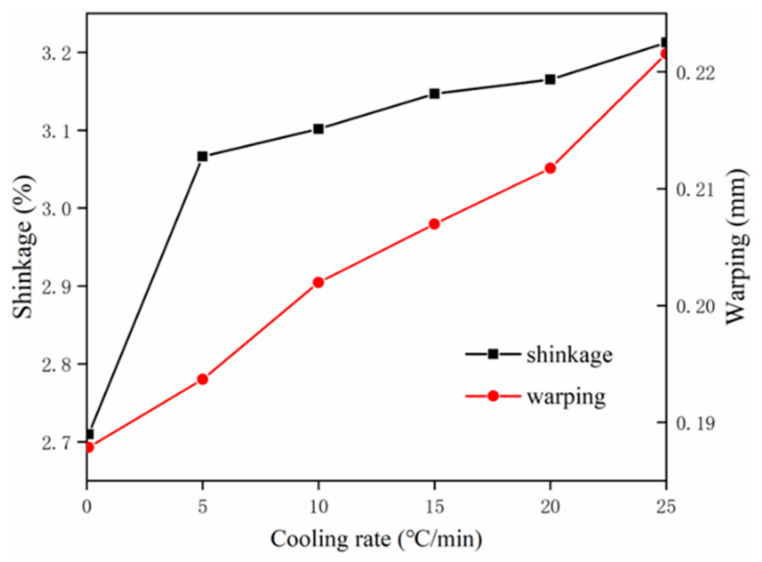
The influence of cooling rate on shrinkage and warping.

**Figure 14 polymers-12-01373-f014:**
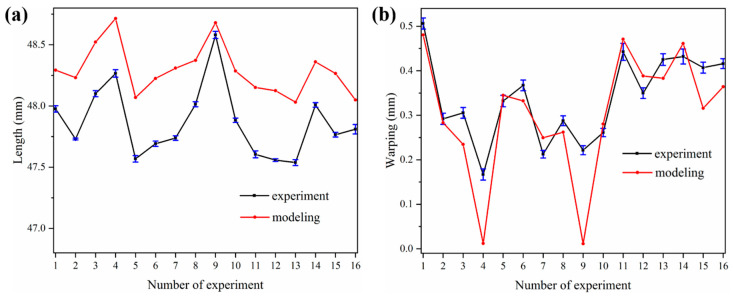
Length and warping respect to the orthogonal design and comparison of experiment and modeling: (**a**) length; (**b**) warping.

**Figure 15 polymers-12-01373-f015:**
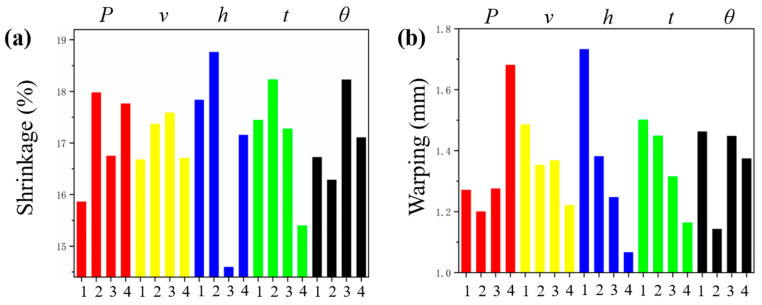
Effect of process parameters on shrinkage and warping: (**a**) experimental shrinkage; (**b**) experimental warping; (**c**) modeling shrinkage; (**d**) modeling warping.

**Figure 16 polymers-12-01373-f016:**
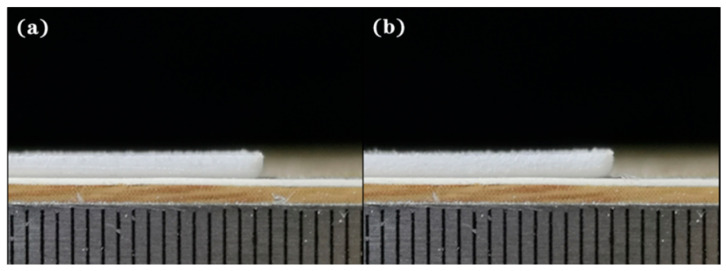
Verification of optimal sets of parameters: (**a**) *P*_2_*v*_4_*h*_4_*t*_4_*θ*_2_; (**b**) *P*_1_*v*_4_*h*_4_*t*_4_*θ*_2_.

**Figure 17 polymers-12-01373-f017:**
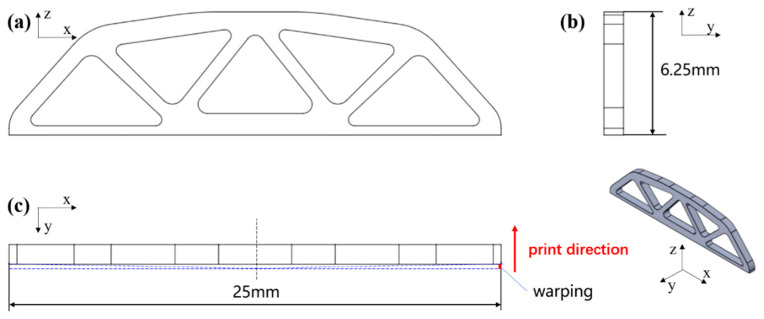
The orthographic sections of the representative structure: (**a**) front view along the *y*-axis; (**b**) left view along the *x*-axis; (**c**) top view along the *z*-axis.

**Figure 18 polymers-12-01373-f018:**
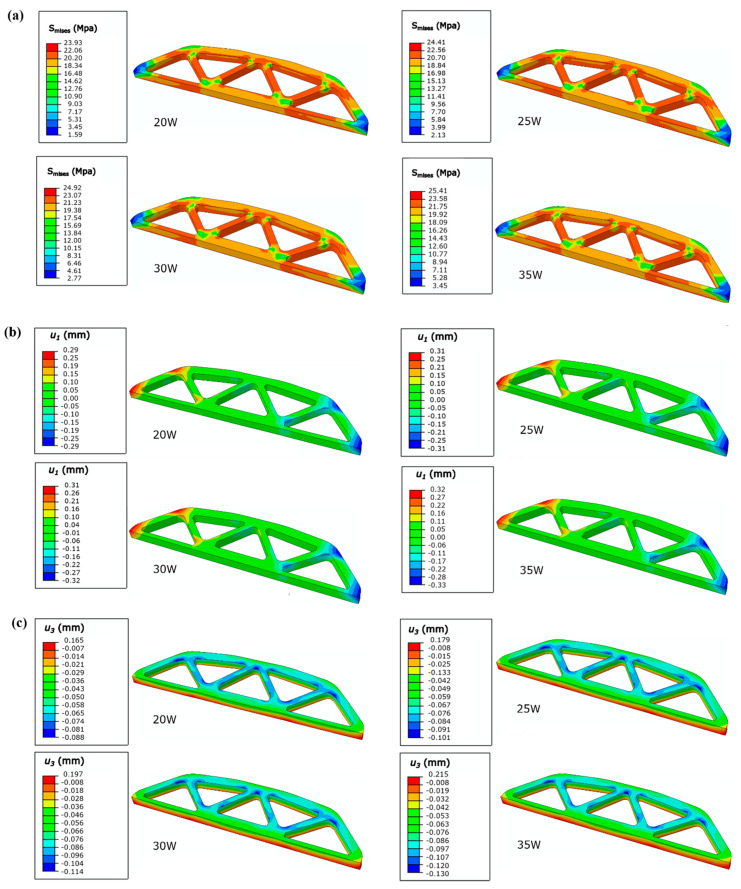
Illustration of (**a**) *S_mises_* distribution; (**b**) *u_1_* distribution; (**c**) *u_3_* distribution of the representative design.

**Figure 19 polymers-12-01373-f019:**
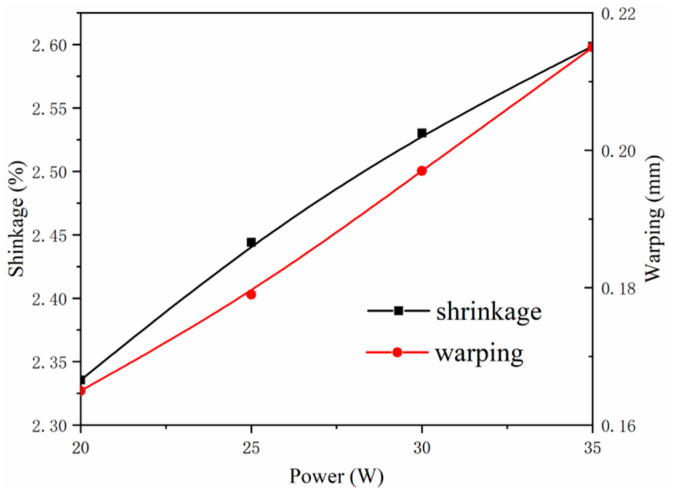
Shrinkage and warping of the representative structure.

**Table 1 polymers-12-01373-t001:** The process conditions and laser constants.

Symbol	Process Parameters	Value
*A*	Power absorption coefficient	0.9
*β*	Extinction coefficient/m^−1^	7500
*r_laser_*	Laser beam diameter/mm	0.6
*T_b_*	Initial bed temperature/°C	170
*T_c_*	Chamber temperature/°C	120

**Table 2 polymers-12-01373-t002:** Specific material properties of PA12; (bulk density *ρ*, melting point *T_m_*, heat distortion temperature *T_h_*, tensile strength *σ_b_*, tensile modulus *E*).

*ρ* (g/cm^3^)	Color	*T_m_* (°C)	*T_h_* (°C)	*σ_b_* (MPa)	*E* (MPa)
1.06	white	183	83.5	46	1602

**Table 3 polymers-12-01373-t003:** Levels of each parameter (laser power *P*, scanning speed *v*, layer thickness *t*, hatching space *h*, and “layer-layer angle” *θ*).

Parameter	Levels of Parameter			
	1	2	3	4
*P* (W)	20	25	30	35
*v* (mm/s)	3000	3500	4000	4500
*h* (mm)	0.15	0.2	0.25	0.3
*t* (mm)	0.1	0.15	0.17	0.19
*θ* (°)	15	45	75	90

**Table 4 polymers-12-01373-t004:** Crystallization enthalpy and maximum crystallinity in different cooling rates.

DSC	Cooling Rate (°C/min)					
	0.8	5	10	15	20	25
*∆H* (J/g)	5.89	39.79	43.26	47.27	51.68	54.08
*λ_max_* (%)	4.40	29.69	32.28	35.28	38.57	40.36

**Table 5 polymers-12-01373-t005:** Analysis of optimal sets of parameters of orthogonal experiment and modeling.

Type	Objective	Optimal Combination of Parameters				
		*P*(W)	*v*(mm/s)	*h*(mm)	*t*(mm)	*θ*(°)
Experiment	Shrinkage	1	1	3	4	2
	R	*h* > *t* > *P* > *θ* > *v*				
Modeling	Shrinkage	1	4	3	4	4
	R	*h* > *t* > *P* > *θ* > *v*				
Experiment	Warping	2	4	4	4	2
	R	*h* > *P* > *t* > *θ* > *v*				
Modeling	Warping	1	4	4	4	2
	R	*t* > *h* > *P* > *θ* > *v*				

**Table 6 polymers-12-01373-t006:** Information of the derived empirical formulas of shrinkage and warping.

Objective	F Value	RMSE *	MFE **	Minimum Parameter Combination				
				*P* (W)	*v* (mm/s)	*h* (mm)	*t* (mm)	*θ* (rad)
*Lε*	33575	0.0002	0.0001	20.4185	4495.7645	0.2647	0.1000	0.2618
*u*	26435	0.0009	0.0005	34.9864	3000.3630	0.1500	0.1900	1.2724

* Root mean squared error ** Maximum of fitting error.

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
