# Peer review of "Numerical Model and Experimental Validation for Laser Sinterable Semi-Crystalline Polymer: Shrinkage and Warping"

_polymers, 2020, doi:10.3390/polym12061373_

Round 1

Reviewer 1 Report

This manuscript reports on a numerical model for selective laser sintering of a semi-crystalline polymer and an experimental study in order to validate it. In selective laser sintering dimensional accuracy may be critically influenced by shrinkage and warping and thus, being able to link the process parameters to the shrinkage and warping taking place and predict them, it is of great importance. In this manuscript, and using a polyamide as model polymer, a theoretical model combined with heat-transfer, thermo-mechanical, and material crystallization kinetics is developed to predict temperature distribution, residual stress, and entire shrinkage/warping in part-scale. The objectives are clear and the idea is interesting. However, there are some mistakes that make difficult to follow the document. For instance, figures numbers are incorrect, tables in the supplementary document are not cited in consecutive order in the main text, symbols (or acronyms) used to refer to shrinkage and warping used in Figure 3, in the main text and in the tables in supplementary information are not the same.

Other questions:

  • How is the energy density calculated: how is the volume determined?
  • Error bars should be added in the graphs corresponding to experimental results
  • What is the reason that for warping in some cases model results are very different than the experimental ones (figure 12 according to the caption/figure 14 according to the text). For samples 4 and 9 while model gives values close to 0, experimental results are around 0.2 mm. It is also difficult to understand why model gives those low values of warping: what is the relevant parameter that allows this?
  • The term “optimal parameters” should be described in a more precise way

Reviewer 2 Report

There are several issues in the manuscript that should be addressed before further consideration for publication.

  1. There has been numerous work that has done simulations for the SLS process. What is the key novelty of this modelling?
  2. For Table 2, are the properties measured or specified?
  3. It is understood that the scanning pattern has an effect on the shrinkage etc of the parts. Is there any specific scan strategy used and considered?- Manshoori Yegeneh et al. (2019), An efficient scanning algorithm for improving accuracy based on minimising part warping in selected laser sintering process, Virtual and Physical Prototyping 14 (1), 59-78
  4. In addition, cubic samples are used for verification. Is there any consideration for other features, geometries etc? There should be some benchmarking parts done
  5. Any verification using other polymers? What are the deviations?

Round 2

Reviewer 1 Report

The authors have adequately addressed my concerns. 

Reviewer 2 Report

NA